# Isolation and Anticancer Progression Evaluation of the Chemical Constituents from *Bridelia balansae* Tutcher

**DOI:** 10.3390/molecules28166165

**Published:** 2023-08-21

**Authors:** Lihan Zhao, Wen-Jian Xie, Yin-Xiao Du, Yi-Xuan Xia, Kang-Lun Liu, Chuen Fai Ku, Zihao Ou, Ming-Zhong Wang, Hong-Jie Zhang

**Affiliations:** 1School of Chinese Medicine, Hong Kong Baptist University, Kowloon, Hong Kong SAR, Chinaliukanglun@hkbu.edu.hk (K.-L.L.);; 2Department of Biology, Georgia State University, Atlanta, GA 30303, USA; 3Department of Materials Science and Engineering, Stanford University, Stanford, CA 94305, USA; 4College of Pharmacy, Shenzhen Technology University, Shenzhen 518118, China

**Keywords:** *Bridelia balansae*, lignan compounds, colorectal cancer, cytotoxicity, apoptosis

## Abstract

The dichloromethane extract of the roots of *Bridelia balansae* Tutcher (Phyllanthaceae) was found to show potential anticancer activity against HCT116 colorectal cancer cell. Our bioassay-guided phytochemical investigation of the roots of *B. balansae* led to the identification of 14 compounds including seven lignans (**1**–**7**), three phenylbenzene derivatives (**8**–**10**), two flavanone (**11**–**12**), and two triterpenoids (**13**–**14**). Among them, 4′-demethyl-4-deoxypodophyllotoxin (**1**) is the first aryltetralin lignan compound identified from this plant species. In addition, the stereochemistry of **1** was validated by X-ray crystallography for the first time, and its distinguished cytotoxic effect on HCT116 cells with an IC_50_ value at 20 nM was induced via an apoptosis induction mechanism. Compound **1** could also significantly decrease the migration rate of HCT116 cells, indicating its potential application against cancer metastasis. The western blot analysis showed that **1** has the potential to inhibit cell proliferation and metastasis. Treatment of **1** resulted in the downregulation of matrix metalloproteinases 2 (MMP2) and p-Akt, while p21 was upregulated. Collectively, the present study on the phytochemical and biological profile of *B. balansae* has determined the plant as a useful source to produce promising anticancer lead compounds.

## 1. Introduction

Natural products, with sources from plants, marine organisms, and microorganisms, have been applied for the treatment of a range of types of diseases in folk medicine since ancient times and played a key role in drug discovery historically [1,2]. With classical natural product chemistry methodologies, a variety of bioactive compounds from natural product resources have been discovered as current drug candidates [1,3]. These therapeutic areas include cardiovascular diseases [4] and multiple sclerosis [5], but cancer and infectious diseases remain the therapeutic areas of which natural products made the major contributions [6,7]. Specifically, over 60% of currently used anticancer agents are derived from natural products, and intensive studies by a random selection screening program funded by the United States National Cancer Institute (NCI) led to the discovery and development of important chemotherapeutics such as the vinca alkaloids, vinblastine, vincristine, and the isolation of the cytotoxic podophyllotoxins [8,9].

The plant genus *Bridelia,* mainly distributed in tropical and subtropical regions of the Africa and Asia, belongs to the Phyllanthaceae family (formerly Euphorbiaceae) with approximately 60 species [10]. Among them, a dozen species of *Bridelia* genus have been used as traditional folk and ethnic medicinal plants in Africa and southeast Asia countries, for the treatment of infective diseases caused by parasitic, bacterial, malarial, viral, oral, or sexual pathogens, glucometabolic and bowel disorders, arthritis [10], and hypertension [11]. They can also be used as analgesics, antianemic purposes, anticonvulsants [10], poison antidotes [12], and antioxidatives [13], or for the purposes of wound healing [12], relieving itching [13], and resolving cough, fever, and jaundice [14]. Consistent with their traditional use, the crude extracts from this plant genus have been reported to have a wide range of bioactivities including potential anticancer activities [15,16]. However, the study of the substance basis of pharmacological effects are rarely involved.

The dichloromethane fraction of the roots of *Bridelia balansae* Tutcher was found to show potential anticancer activity, with an IC_50_ value of 0.79 μg/mL against human colorectal carcinoma cells HCT116. According to APG III System, *B. balansae* belongs to the *Phyllanthaceae* family, and could be found mainly in Lingnan area of China, including Fujian, Guangdong, Guangxi, Guizhou, Hainan, Sichuan, Taiwan, and Yunnan Province, together with other countries in Asia such as Japan (Ryukyu Islands), Laos, and Vietnam. It has been used as a folk medicine to treat gastropathy and glometulonephropathy in China [17], but there is no prior report of the bioactivity of the chemical constituents from this plant. Thus, we are encouraged to carry out further phytochemical studies to discover bioactive compounds from this plant based on the potent bioactivity of the plant extract. Herein, we describe the isolation, structural identification, and biological activity evaluation of the compounds obtained from *B. balansae*.

## 2. Results

### 2.1. Compound Isolation and Structure Identification

Compounds **1**–**14** were identified as 4′-demethyl-4-deoxypodophyllotoxin (**1**) (Appendix A) [18,19], deoxypodophyllotoxin (**2**) [20], polygamain (**3**) [21], yatein (**4**) [22], pinoresinol (**5**) [23], medioresinol (**6**) [24], (+)-Syringaresinol (**7**) [25], gallic acid (**8**) [26], gallicin (**9**) [27], methyl-3,5-dimethoxy-4-hydroxybenzoate(**10**) [28], gallocatechin (**11**) [29], and epigallocatechin gallate (**12**) [30], epifriedelanol (**13**) [31], and friedelin (**14**) [31] (Figure 1) by comparing the NMR and HRMS data reported from previous studies. The prior absolute stereochemistry of **1** was only established by using NMR method since it was first discovered in 1984, and no X-ray crystallographic data have been reported so far to validate the absolute configuration [18]. Considering the chirality diversity of the carbon skeleton of aryltetralin lignans, we nurtured the fine crystals of **1** from acetone to obtain high-quality X-ray crystallographic data with CuKα radiation. The resulted Flack parameter of 0.13 (17) from the refinement of the X-ray data, together with the comparison of the reported optical rotation data of podophyllotoxin [32], confirmed the absolute stereochemistry of **1** (Figure 2). In addition, we measured the melting point (248.5–249.0 °C) and the optical rotation ([α]^20^_D_ = −96°, *c* = 0.3, CHCl_3_) of **1**, which are consistent with the literature report [19]. The isolated and purified compounds are stored at −80 °C and then dissolved into DMSO for treatment. Dissolved compounds were stored at −20 °C. Through a bioassay-guided phytochemical investigation of the roots of *B. balansae*, we were able to identify seven lignans, three phenylbenzene derivatives, two flavanone, and two triterpenoids. The newly isolated compounds were tested for their anticancer activity potential against HCT116 colorectal cancer cells. We subsequently investigated the mechanism investigation of **1**.

### 2.2. Cytotoxic Effect of Isolated Compounds against HCT116 Cancer Cells

The in vitro cytotoxicity of each isolated compound at 20 μg/mL was evaluated against the human colorectal carcinoma HCT116 cells (Table 1). Among the tested compounds, **1**, **8**, and **9** were the most active ones with an inhibition rate larger than 50% at 20 μg/mL, and thus further tested for their IC_50_ values (Table 2). Among these three compounds, only **1** showed potent cytotoxicity against HCT116 cells with an IC_50_ value of 0.02 μM, and thus was chosen for further mechanism studies.

### 2.3. Compound ***1*** Induced Apoptosis in HCT116 Cancer Cells

It has been previously reported that 4′-demethyl-4-deoxypodophyllotoxin (**1**) could induce cell cycle arrest with a significantly increased G2/M peak after treatment of HCT-116 cells [33]. We further investigated whether the antiproliferative activities of **1** against HCT116 cancer cells were associated with apoptosis, which is the most common mechanism of action of anticancer agents. As shown in Figure 3, the percentages of apoptotic cells were analyzed by flow cytometric analysis. While the percentages of apoptotic cells did not change significantly when **1** was applied at concentrations lower than its IC_50_ value, the number of apoptotic cells was remarkably increased when **1** was applied at 26 nM (slightly higher than its IC_50_ value). These findings suggest that the anti-proliferative effect of **1** in HCT116 cells could be derived from apoptosis.

### 2.4. Compound ***1*** Reduced Cellular Mobilization of HCT116 Cancer Cells

To fully elucidate the anticancer properties and examine whether compound **1** also affects cellular mobilization of HCT116 cancer cells, scratch wound assay was utilized after applying indicated concentrations of **1**. In Figure 4, we observed that the treatment with **1** at 3.25 and 6.5 nM significantly inhibited the mobilization of HCT116 cells in a dose-dependent manner, whose healing effects were obvious after 24 h. We thus conclude that the healing effects of HCT116 cells have been inhibited when treated with **1** from 0 to 6.5 nM in vitro.

### 2.5. Compound ***1*** Disrupted Microtubules of HCT116 Cancer Cells

As compound **1** shared similar chemical structures with podophyllotoxin, which is known as a microtubule-destabilizing agent, we further evaluated the effects of **1** on microtubule dynamics by immunofluorescent staining. As shown in Figure 5, HCT116 cells exhibited disruption of the microtubule network after the treatment of **1** for 48 h, especially at concentrations of 13 and 26 nM. Thus, the results confirmed our hypothesis that compound **1** has the ability to disrupt the microtubule assembly, indicating that microtubule dynamics may be one of the targets of the anti-proliferation and anti-migration effects of **1**.

### 2.6. Effects of Compound ***1*** on Apoptosis and Migration Regulatory Proteins of HCT116 Cancer Cells

To test the effects of compound **1** on apoptosis and migration regulatory proteins in HCT116 cancer cells, we performed western blotting analysis. Figure 6 showed that **1** could decrease the protein expression of pAKT and MMP2, while increasing the protein expression of p21. These regulations are all dose dependent, and the effects are most significant at the concentration of a 13 nM treatment. Collectively, our results suggest that the antitumor effect of **1** is majorly associated with the p21-mediated apoptosis and MMP2-mediated migration, which are both regulated by their upstream, phosphorylated AKT.

## 3. Discussion

In previous studies, the investigations on *Bridelia* plant genus mainly focused on the validation of the folk or ethnic medical application using the crude extracts [10,11,12,13]. In the present study, we in-depth investigated the major phytochemical profile of the roots of *Bridelia balansae* by bioassay guided separation. The 14 compounds that were identified belong to aryltetralin [18,19,20,21,22] and 2-aryltetrahydrofuran [23,24,25] lignans, organic tannic acids [26,27,28], flavanones [29,30], and triterpenoids [31]. Although these compounds are known, these findings expanded the diversity of the plant sources of these bioactive compounds. More importantly, the absolute configuration of the lactone ring is difficult to confirm by the regular HRMS and NMR spectroscopies in terms of aryltetralin skeletons in some extreme cases. The confirmation of the absolute structure of compound **1** by X-ray crystallography method for the first time is beneficial to the structural determination of future unknown natural analogues. In addition, flavonoid–tannin conjugates are a rare subtype of small molecules in nature. As an important precursor of podophyllotoxin, aryltetralin lignans also exhibited a pronounced biological activity mainly as anti-tumor agents. Considering the synthetic challenges due to multiple chiral centers, natural aryltetralin lignans isolated from the roots of *Bridelia balansae* can partially overcome the shortage of chemical sources, which will provide the substantial support of further pharmacological studies.

In the present study, we also evaluated the bioactivity of compound **1** in depth, including the elucidation of its anti-proliferative property, and its induction of apoptosis and migration using cellular platforms. Previously, it has been reported that 4′-demethyl-4-deoxypodophyllotoxin (**1**) could induce cell cycle arrest with a significantly increased G2/M peak after treatment of HCT-116 cells [33]. Cell cycle arrest could either provide cancer cells the opportunity for DNA damage repair, or lead to cell death [34]. Among all the cell death pathways, apoptosis is one of the most common mechanisms to eliminate damaged cells that will further restrict tumorigenesis [35]. The data of our flow cytometric analysis show that the anti-proliferative effect of **1** on human colorectal carcinoma cells is primarily derived from apoptosis. The apoptosis is mediated by the upregulation of p21 and might be caused by the effects of microtubule disassembly of **1**. This result is consistent with podophyllotoxin, an aryltetralin lignan congener of **1** and a well-known microtubule-destabilizing agent [36,37]. In the present study, compound **1** was found to induce a significant apoptosis event. Thus, further research is needed to determine if the aryltetralin lignan-induced apoptotic pathway is linked to the mechanism of action of the microtubule-destabilizing agents, such as the clinically used anticancer drugs etoposide and teniposide, which are derived from podophyllotoxin. Notably, it has been reported that targeted microtubule disruption can selectively inhibit metastasis, although these therapies are currently focusing exclusively on tumor growth [38]. Therefore, the anti-migration effects of **1** might also be contributed to by its microtubule disruption effects. These effects collectively regulated MMP2, and thus induced the reduction of cell migration. We also noticed a significant dose-dependent decrease of the expression of phosphorylated AKT, which is the upstream of both p21 and MMP2. It has been well acknowledged that AKT signaling promotes tumor cell proliferation, growth, and metastasis by activating its downstream effectors, and thus the inhibitors targeting this signaling pathway have been considered as the most effective treatment strategy for cancer [39]. The wound healing (or scratch) assay is a method to measure two-dimensional cell migration [40]. It involves creating an artificial gap in a confluent cell monolayer at 0 h, and tracking the movement of HCT116 cells through microscopy imaging at both 0 and 24 h. The narrowing of the gap indicated that compound **1** was capable of inhibiting the migration of HCT116 cells. Collectively, our study provides a basis for the anti-cancer effects of compound **1** against HCT116 cancer cells, which might lead to further structural modification of its functional groups or more detailed investigations of its mechanism, serving as a potential lead compound in colon cancer treatment.

## 4. Materials and Methods

### 4.1. General

Flash silica gel (300 mesh, Qingdao Haiyang Chemical Co., Ltd., Qingdao, China) was used as a stationary phase in column chromatography. Thin-layer chromatography (TLC) (Qingdao Haiyang Chemical Co., Ltd., Qingdao, China) was used to monitor the fractionation progress. Analytical and semi-preparative HPLC were performed on an Agilent Technologies Series 1100 HPLC with a diode array detector (DAD, Agilent Technologies, Inc., Santa Clara, CA, USA) equipped with a semi-preparative Thermo-C18 (150 × 4.6 mm) column or Nawei-UniSil 10–120 C18 Ultra (250 × 21.2 mm) column. A Bruker Ascend 400 MHz spectrometer with standard parameters supplied by the vendor (Bruker, Karlsruhe, Germany) was applied to measure one-dimensional (1D) and two-dimensional (2D) NMR spectra. High-resolution mass spectrometry was performed on an Agilent QTOF-6542 (Santa Clara, CA, USA). X-ray crystallographic data were obtained on a Bruker D8 Venture X-Ray Diffractometer (Karlsruhe, Germany). The optical rotation of a compound was measured on a polarimeter (JACSCO, P1010, Tokyo, Japan). The melting point was detected by a melting point tester (Electrothermal, London, UK). Mili Q water and acetonitrile (ACN, HPLC grade, Duksan, Republic of Korea) were used as a mobile phase of HPLC. Methanol (HPLC grade, Duksan, Republic of Korea), ethyl acetate (EA, HPLC grade, Duksan, Republic of Korea), petroleum ether (PE, HPLC grade, Duksan, Republic of Korea), and dichloromethane (DCM, HPLC grade, Duksan, Republic of Korea) were used for the fractionation and isolation of compounds.

### 4.2. Plant Material

The roots of *B. balansae* were collected from Jianfengling National Nature Reserve, Hainan Province in 1 August 2014. The collected plant materials were authenticated by Prof. CHEN Hubiao from School of Chinese Medicine, Hong Kong Baptist University. A voucher specimen (No. SHABB20140801) was deposited at School of Chinese Medicine, Hong Kong Baptist University, Hong Kong, China.

### 4.3. Compound Isolation and Structure Identification

The dry weight of the root part was 5.0 kg. The root part of plant material was then ground and extracted with 50 L methanol at room temperature to afford 1.0 kg crude extract, which was partitioned with ethyl acetate (EA) 3 times to produce an EA extract (300 g). The EA extract was further fractionated on a silica gel column, eluted with petroleum ether (PE), PE/DCM (1:1), DCM/methanol (MeOH) (1:0–0:1) gradient to afford 6 fractions (Fractions 1–6) (Appendix A). Compounds **2** (20 mg), **9** (3 mg), **13** (3 mg), and **14** (10 mg) were isolated from Fraction 3 by using HPLC with an isocratic elution (55% ACN: 45% H_2_O as mobile phase, 3 mL/min) (Appendix A). Compounds **1** (30 mg), **3** (5 mg), **4** (6 mg), **5** (3 mg), **6** (2 mg), **7** (3 mg), **8** (2 mg), **10** (2 mg), and **11** (6 mg) were isolated from the major active Fraction 4 by using HPLC with an isocratic elution (50% ACN: 50% H_2_O as mobile phase, 3 mL/min) (Appendix A), and compound **12** (7 mg) was isolated from Fraction 5 (Appendix A).

Compound **1**. Colourless crystal (acetone); mp 248.5–249.0 °C; [α]^20^_D_ = −96° (*c* 0.3, CHCl_3_); ^1^H NMR (400 MHz, CDCl_3_) *δ*: 6.65 (s, 1H), 6.51 (s, 1H), 6.34 (s, 2H), 5.93 (dd, *J* = 10.4, 1.2, 2H), 5.42 (brs, 1H), 4.58 (d, *J* = 2.8, 1H), 4.41–4.44 (m, 1H), 3.88–3.92 (m, 1H), 3.77 (s, 6H), 3.04–3.07 (m, 1H), 2.70–2.77 (m, 3H). ^13^C NMR (100 MHz, CDCl_3_): *δ*: 175.0, 146.9, 146.7, 146.4, 133.8, 131.8, 130.8, 128.3, 110.5, 108.5, 107.9, 101.2, 72.1, 56.4, 47.6, 43.6, 33.1, 32.7. HRESIMS *m*/*z:* 385.1287 [M + H]^+^; calcd for C_21_H_21_O_7_ 385.1282 [18].

Compound **2**. White powder; ^1^H NMR (400 MHz, CDCl_3_) *δ*: 6.66 (s, 1H), 6.51 (s, 1H), 6.34 (s, 2H), 5.92 (dd, *J* = 9.2, 1.2, 2H), 4.59 (d, *J* = 2.8, 1H), 4.43–4.47 (m, 1H), 3.89–3.94 (m, 1H), 3.79 (s, 3H), 3.74 (s, 6H), 3.03–3.10 (m, 1H), 2.69–2.80 (m, 3H). ^13^C NMR (100 MHz, CDCl_3_): *δ*: 174.9, 152.5, 147.0, 146.7, 136.9, 136.3, 130.6, 128.3, 110.5, 108.5, 108.2, 101.2, 72.1, 60.8, 56.2, 47.5, 43.7, 33.1, 32.7. HRESIMS *m*/*z*: 399.1444 [M + H]^+^; calcd for C_22_H_23_O_7_ 399.1438 [19].

Compound **3.** White powder; ^1^H NMR (400 MHz, CDCl_3_) δ: 6.66 (d, *J* = 7.2, 2H), 6.60 (d, *J* = 7.6, 2H), 6.47 (s, 1H), 5.92 (s, 2H), 5.89 (dd, *J* = 6.0, 1.2z, 2H), 4.56 (d, *J* = 4.4, 1H), 4.42–4.46 (m, 1H), 3.92 (t, *J* = 8.0, 1H), 3.06 (d, *J* = 10.4, 1H), 2.71–2.77 (m, 3H). ^13^C NMR (100 MHz, CDCl_3_): *δ*: 174.8, 147.2, 146.9, 146.8, 146.5, 134.5, 131.0, 128.2, 124.2, 111.1, 110.4, 108.5, 107.7, 101.2, 100.9, 72.1, 47.3, 43.2, 33.1, 32.6. HRESIMS *m*/*z*: 353.1021 [M + H]^+^; calcd for C_20_H_17_O_6_ 353.1020 [20].

Compound **4**. White powder; ^1^H NMR (400 MHz, CDCl_3_) *δ*: 6.69 (d, *J* = 7.2 Hz, 1H), 6.46 (d, *J* = 7.6 Hz, 1H), 6.35 (s, 2H), 6.19 (s, 1H), 5.93 (dd, *J* = 4.0, 1.2 Hz, 2H), 4.17 (dd, *J* = 9.2, 7.2, 1H), 3.85–3.89 (m, 1H), 3.74–3.83 (brs, 9H), 2.89–2.91 (m, 1H), 2.50–2.60 (m, 4H). ^13^C NMR (100 MHz, CDCl_3_): *δ*: 178.6, 153.3, 147.9, 146.4, 133.3, 131.5, 121.5, 108.8, 108.3, 106.2, 105.5, 101.1, 71.2, 60.9, 56.1, 46.5, 41.0, 38.4, 35.3. HRESIMS *m*/*z*: 401.1599 [M + H]^+^; calcd for C_22_H_25_O_7_ 401.1595 [21].

Compound **5**. White powder; ^1^H NMR (400 MHz, CDCl_3_) δ: 6.88–6.90 (m, 4H), 6.81 (dd, *J* = 2.0, 8.0 Hz, 2H), 5.62 (brs, 2H), 4.73 (d, *J* = 4.4 Hz, 2H), 4.22–4.26 (m, 2H), 3.89 (brs, 6H), 3.85 (dd, J = 3.6, 9.2 Hz, 2H), 3.08–3.11 (m, 2H). ^13^C NMR (100 MHz, CDCl_3_): 146.7 (2C), 145.2 (2C), 132.9 (2C), 118.9 (2C), 114.3 (2C), 108.6 (2C), 85.9 (2C), 71.7 (2C), 55.9 (2C), 54.1 (2C. H RESIMS *m*/*z*: 359.1493 [M + H]^+^; calcd for C_20_H_23_O_6_ 359.1489 [22].

Compound **6**. White powder; ^1^H NMR (400 MHz, CDCl_3_) *δ*: 6.81–6.83 (m, 2H), 6.76 (dd, *J* = 1.2, 8.0 Hz, 1H), 6.51 (s, 2H), 4.65 (dd, *J* = 3.6, 7.6 Hz, 2H), 4.16–4.21 (m, 2H), 3.80–3.85 (brs, 11H), 3.03 (brs, 2H). ^13^C NMR (100 MHz, CDCl_3_): 147.2 (2C), 146.7, 145.2, 134.4, 132.9, 132.1, 118.9, 114.3, 108.6, 102.7 (2C), 86.1, 85.8, 71.8, 71.6, 56.4 (2C), 55.9, 54.4, 54.1. HRESIMS *m*/*z*: 389.1593 [M + H]^+^; calcd for C_21_H_25_O_7_ 389.1595 [23].

Compound **7**. White powder; ^1^H NMR (400 MHz, CDCl_3_) *δ*: 6.59 (brs, 4H, Ar-H), 5.60 (brs, 2H), 4.73 (d, *J* = 3.6 Hz, 2H), 4.28 (t, *J* = 6.0, 2H), 3.82–3.92 (brs, 14H), 3.10 (brs, 2H). ^13^C NMR (100 MHz, CDCl_3_): 147.2 (4C), 134.3 (2C), 132.0 (2C), 102.7 (4C), 86.0 (2C), 71.8 (2C), 56.4 (4C), 54.3 (2C). HRESIMS *m/z*: 419.1707 [M + H]^+^; calcd for C_22_H_27_O_8_ 419.1700 [24]. 

Compound **8**. Brown powder; ^1^H NMR (400 MHz, acetone-*d*_6_) *δ*: 7.06 (s, 2H). ^13^C NMR (100 MHz, CDCl_3_): *δ*: 170.9, 146.4 (2C), 139.4, 122.7, 110.3 (2C). HRESIMS *m*/*z*: 171.0280 [M + H]^+^; calcd for C_7_H_7_O_5_ 171.0288 [25].

Compound **9**. Brown powder; ^1^H NMR (400 MHz, acetone-*d*_6_) *δ*: 7.07 (brs, 2H), 3.83 (s, 3H). ^13^C NMR (100 MHz, acetone-*d*6): *δ*: 169.1, 146.5, 139.8, 121.5, 110.1, 52.4. HRESIMS *m*/*z*: 185.0447 [M + H]^+^; calcd for C_8_H_9_O_5_ 185.0444 [26].

Compound **10**. Brown powder; ^1^H NMR (400 MHz, acetone-*d*_6_) *δ*: 7.32 (s, 2H), 3.94 (s, 6H), 3.89 (s, 3H). ^13^C NMR (100 MHz, CDCl_3_): 166.8, 146.6, 139.2, 127.1, 106.6, 56.4, 52.1. HRESIMS *m/z*: 213.0781 [M + H]^+^; calcd for C_10_H_13_O_5_ 213.0757 [27].

Compound **11**. White powder; ^1^H NMR (400 MHz, MeOD) *δ*: 6.41 (s, 2H), 5.93 (d, *J* = 2.0, 1H), 5.87 (d, *J* = 2.4, 1H), 4.54 (d, *J* = 7.2, 1H), 3.97 (dd, *J* = 12.8, 7.6 Hz, 1H), 3.30–3.31 (m, 1H), 2.80 (dd, *J* = 16.0, 5.6, 1H), 2.50 (dd, *J* = 16.0, 7.6, 1H). ^13^C NMR (100 MHz, CDCl_3_): *δ*: 157.8, 157.6, 156.8, 146.9 (2C), 134.0, 131.6, 107.2 (2C), 100.8, 96.3, 95.6, 82.9, 68.8, 28.1. HRESIMS *m*/*z*: 307.0809 [M + H]^+^; calcd for C_15_H_15_O_7_ 307.0812 [28].

Compound **12**. White powder; ^1^H NMR (400 MHz, CDCl_3_) *δ*: 6.95 (s, 2H), 6.50 (s, 2H), 5.96 (s, 2H), 5.52 (brs, 1H), 4.97 (s, 2H), 2.96 (dd, *J* = 17.2, 4.4, 1H), 2.82 (dd, *J* = 17.2, 2.4, 1H). ^13^C NMR (100 MHz, CDCl_3_): *δ*: 167.7, 157.9, 157.8, 157.3, 146.7 (2C), 146.3 (2C), 139.8, 133.8, 130.8, 121.5, 110.3 (2C), 106.9 (2C), 99.5, 96.6, 95.9, 78.6, 69.9, 26.9. HRESIMS *m*/*z*: 459.0920 [M + H]^+^; calcd for C_22_H_19_O_11_ 459.0922 [29]. 

Compound **13**. White solid; ^1^H NMR (400 MHz, CDCl_3_) *δ*: 3.73 (brs, 1H), 2.35–2.45 (m, 1H), 2.30–2.35 (m, 1H), 2.20–2.30 (m, 1H), 1.93–2.00 (m, 1H), 1.87–1.92 (m, 1H), 1.70–1.78 (m, 2H), 1.65–1.70 (m, 1H), 1.43–1.50 (m, 6H), 1.32–1.42 (m, 10H), 1.24–1.32 (m, 8H), 1.20–1.24 (m, 1H), 1.15 (d, *J* = 4.4 Hz, 6H), 1.05 (s, 3H), 0.98–1.03 (m, 6H), 0.92–0.97 (m, 6H), 0.84–0.89 (m, 6H), 0.72 (s, 3H). HRESIMS *m*/*z*: 439.4089 [M + H]^+^; calcd for C_30_H_53_O 429.4091 [30].

Compound **14**. White solid; ^1^H NMR (400 MHz, CDCl_3_) *δ*: 2.36–2.42 (m, 1H), 2.27–2.34 (m, 1H), 2.22–2.27 (m, 1H), 1.93–2.00 (m, 1H), 1.72–1.78 (m, 1H), 1.62–1.71 (m, 1H), 1.52–1.58 (m, 5H), 1.43–1.53 (m, 5H), 1.32–1.42 (m, 6H), 1.23–1.32 (m, 3H), 1.20–1.23 (m, 1H), 1.18 (s, 3H), 1.05 (s, 3H), 1.01 (s, 3H), 0.99 (s, 3H), 0.95 (s, 3H), 0.88 (d, *J* = 6.8 Hz, 3H), 0.87 (s, 3H), 0.72 (s, 3H). ^13^C NMR (100 MHz, CDCl_3_): 213.3, 59.5, 58.2, 53.1, 42.8, 42.2, 41.6, 41.3, 39.7, 39.3, 38.3, 37.5, 36.0, 35.6, 35.4, 35.0, 32.8, 32.4, 31.8, 30.5, 30.0, 28.2, 22.3. HRESIMS *m*/*z*: 427.3931 [M + H]^+^; calcd for C_30_H_51_O 427.3934 [30].

### 4.4. X-ray Crystallographic Data of ***1***

Crystals of **1** were obtained from acetone solvent at room temperature. Single-crystal X-ray crystallographic analyses of **1** were obtained on a Bruker D8 Venture X-ray Diffractometer (Karlsruhe, Germany).

Crystallographic data of **1**. C_21_H_20_O_7_, *M* = 384.37, a = 7.8438(7) Å, b = 9.4648(9) Å, c = 12.1581(12) Å; α = 93.206(3)°, β = 91.396(3)°, γ = 90.214(3), V = 900.93(15) Å^3^, T = 297(2) K, space group P1, Z = 2, μ (Cu Kα) = 0.894 mm^−1^; 18,230 reflections collected, 6148 independent reflections (R_int_ = 0.1127). The final R1 value was 0.0940 [I > 2σ(I)]. The final wR (F2) value was 0.2233 [I > 2σ (I)]. The final R1 value was 0.1209 (all data). The final wR (F2) value was 0.2710 (all data). The goodness of fit on F2 was 1.141. Flack parameter = 0.13(17). The crystal structure of **1** has been deposited in the CCDC database and the deposit number is CCDC 2287689. The DOI link can be accessed through DOI: 10.5517/ccdc.csd.cc2gsjf7 (accessed on 9 August 2023).

### 4.5. Cell Proliferation Assay

The human colon cancer cell line HCT116 was purchased from the American Type Culture Collection (Manassas, VA, USA) and maintained in Dulbecco’s modified Eagle’s medium (DMEM), supplemented with 10% fetal bovine serum (FBS) at 37 °C, 5% CO_2_. The effects of isolated compounds on [17] colon cancer cell viability were determined by sulforhodamine B (SRB) assay [41]. The cells were seeded at a density of 5000 cells/well in 96-well microculture plates, in the absence or presence of different concentrations of a compound, incubated for 48 h. Thereafter, 50 µL of cold 50% trichloroacetic acid was added to each well to fix proteins and incubated at 4 °C for at least 1 h. The plate was washed with tap water four times and dried. For each well, 100 µL 0.4% SRB in 1% acetic acid was added to do the staining and incubated for 10 min at room temperature. After the SRB was discarded in the sink, the plate was washed by 1% acetic acid four times and dried. For the optical density (OD) value reading, 200 µL of 10 mM tris base (pH 10) was added to each well and shaken for 30 min or more. The optical absorbance at wavelength 515 nm was detected by a microplate spectrophotometer (BIO-RAD, Benchmark Plus). Percentage growth inhibition was calculated as [OD (cells + samples) − OD (Day 0 cells)]/[OD (cells + 10% DMSO) − OD (Day 0 cells)] = % survival, cytotoxicity = 1 − % survival. The IC_50_ values of the results were analyzed using GraphPad Prism version 8 (GraphPad Software, San Diego, CA, USA). Paclitaxel was used as the positive reference.

### 4.6. Flow Cytometry

The cells were seeded in a 6-well plate (250,000 cells) and incubated for 24 h at 37 °C with 5% CO_2_. After settling, cells were further incubated in the absence or presence of **1** at the concentrations of 6.5, 13, and 26 nM to induce apoptosis for 48 h. The cells were harvested after the incubation period and washed in cold phosphate-buffered saline. The washed cell pellets were resuspended in 100 μL 1× annexin-binding buffer per assay, with 5 μL of FITC annexin V and 1 μL of the 100 μg/mL PI working solution added. The cells were incubated at room temperature for 15 min. After the incubation period, 400 μL of 1× annexin-binding buffer was added and mixed gently on ice. The stained cells were then analyzed by FACS Calibur™ (BD Biosciences, San Jose, CA, USA).

### 4.7. Wound Healing Assay

HCT116 cells were seeded in 6-well plates and cultured with full medium with or without compound **1** treatment until the confluent before the experiment was started. The seeding density was 300,000 cells with 2 mL medium per well. A 10 µL pipette tip was used to make a straight scratch, simulating a wound. After the scratch, the medium was changed into DMEM without FBS.

The wound edges were allowed to be imaged using a 5× objective and were focused on using the focus knob on the microscope. The positions desired to take were selected and imaged by a microscope (Leica DMIRB). After 24 h, the wound edges were imaged again with the same selected positions.

### 4.8. Immunofluoresent Staining

HCT116 cells were seeded in 6-well plates on slides and cultured with full medium with or without compound **1** at different concentrations for 48 h. The seeding density was 300,000 cells with 2 mL medium per well. After fixation in ice-cold acetone: MeOH (1:1, *v*/*v*) for 20 min, cells were incubated with anti-α-tubulin antibody followed by fluorescent-conjugated secondary antibody, visualized, and imaged by a microscope (Leica AF6000, Hong Kong SAR, China).

### 4.9. Western Blot Analysis

Proteins were extracted from HCT-116 cells using an ice-cold lysis buffer containing protease and phosphatase inhibitors (Thermo Fisher Scientific, Waltham, MA, USA). Cell lysates were loaded to 10% SDS-polyacrylamide gel for electrophoresis separation and transferred onto polyvinylidene fluoride membranes by wet electroblotting. The membranes were blocked with 5% non-fat dry milk in Tris-buffered saline containing 0.1% Tween 20 for 1 h at room temperature, followed by incubation with primary antibodies overnight at 4 °C, including p21 (Abcam, Cambridge, UK), MMP2 (CST), GAPDH (Bio-Rad, Herkleys, CA, USA). Subsequently, the membranes were incubated with secondary antibodies of interest for 1 h at room temperature and further visualized.

## 5. Conclusions

Our bioassay-guided phytochemical investigation of the roots of *B. balansae* led to the identification of 14 compounds. Among them, **3** has not been reported in nature previously, and **1** is the first aryltetralin lignan compound identified from this plant species. In addition, the absolute configuration of **1** was validated by X-ray crystallography for the first time, and its distinguished cytotoxic effect on HCT116 cells with an IC_50_ value at 20 nM was induced via an apoptosis induction mechanism. Compound **1** could also significantly decrease the migration rate of HCT116 cells, indicating its potential application against cancer metastasis. Collectively, the present study on the phytochemical and biological profile of *B. balansae* has determined the plant as a useful source to produce promising anticancer lead compounds.

## Figures and Tables

**Figure 1 molecules-28-06165-f001:**
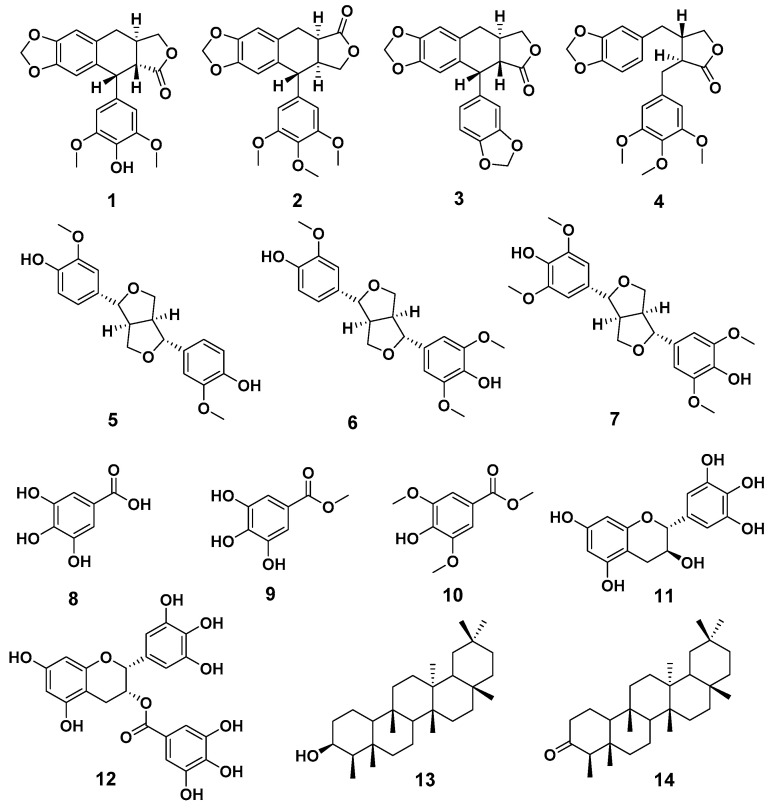
The chemical structures of compounds **1**–**14**.

**Figure 2 molecules-28-06165-f002:**
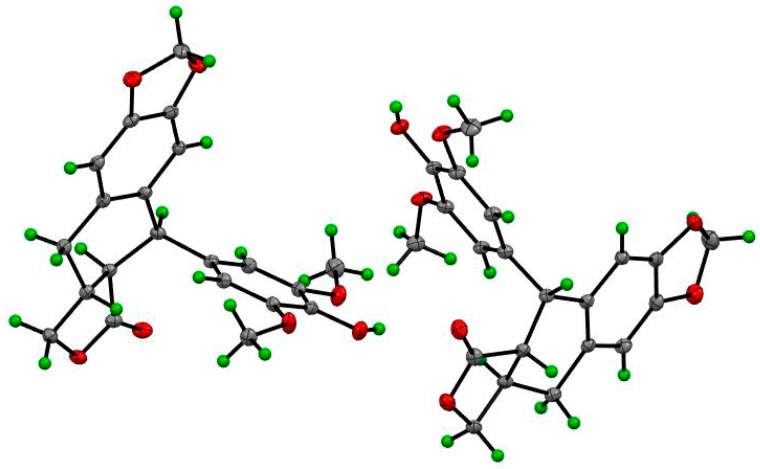
The ORTEP drawing of compound **1**.

**Figure 3 molecules-28-06165-f003:**
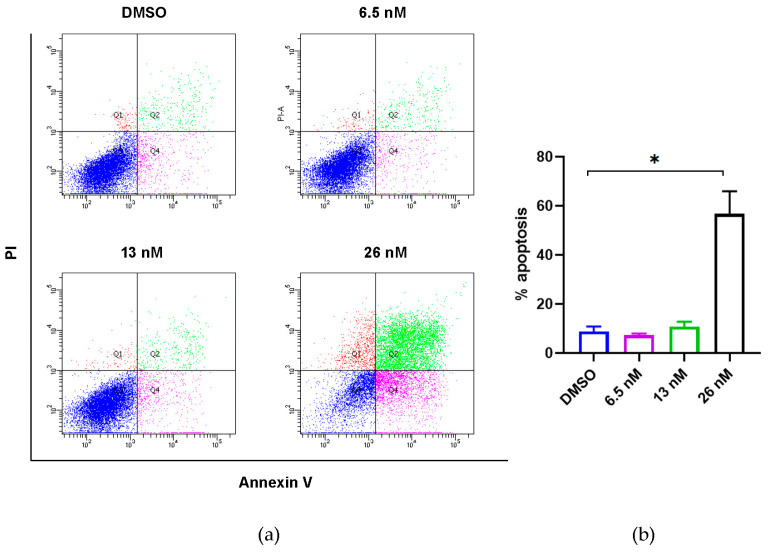
Effects of compound **1** on induction of apoptosis of HCT116 cancer cells. (**a**) Flow cytometric analysis of HCT116 cancer cells using Annexin V/PI staining. Cells were treated with **1** at the indicated concentrations for 48 h prior to labelling with annexin V and PI. Q1 shows the percentage of necrotic cells, and Q2 and Q4 represent late and early apoptotic cells, respectively. Q3 shows the population of normal cells. (**b**) Graphical representation of the percentage of apoptotic cells (sum of Q2 and Q4) from three independent replicates’ data as the mean ± SEM; * *p* < 0.05.

**Figure 4 molecules-28-06165-f004:**
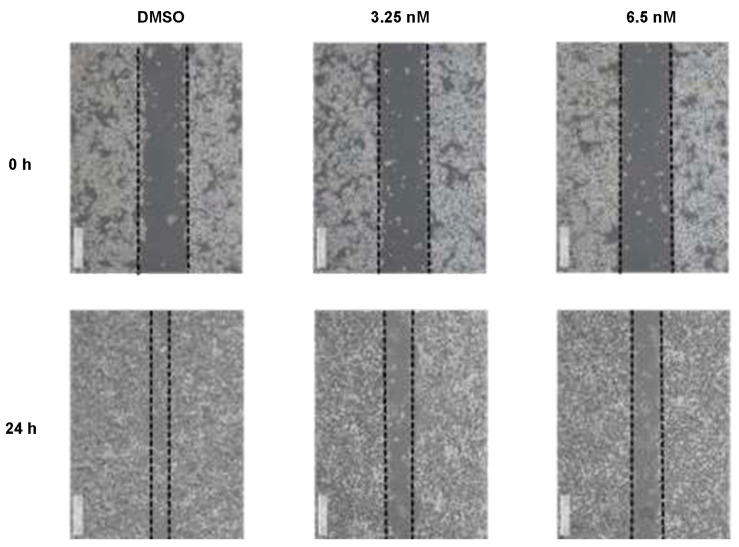
Compound **1** reduced cellular mobilization in HCT116 cells treated with various concentrations of **1**. Images were acquired at 0, 24 h after scratching. The dotted lines define the areas lacking cells. Data were collected from 3 independent experiments and only presentative pictures are shown.

**Figure 5 molecules-28-06165-f005:**
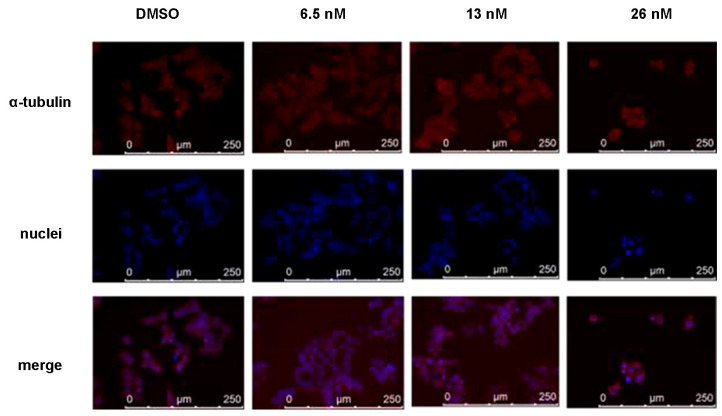
Immunofluorescence study of disruption of the microtubule network in HCT116 cells. Microtubules were visualized with an anti-α-tubulin antibody (red). The nuclei of cells were stained with 4′, 6-diamidino-2-phenylindole and thus showed a blue color.

**Figure 6 molecules-28-06165-f006:**
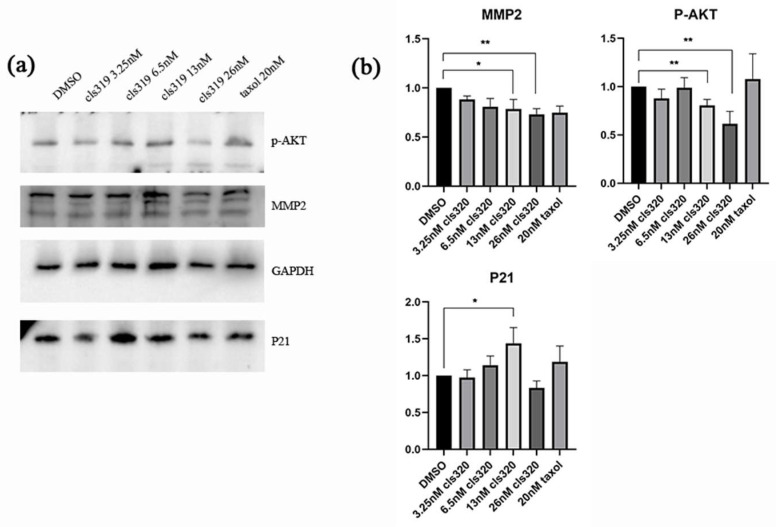
Western blot analysis shows effects of compound **1** on apoptosis and migration regulatory proteins in HCT116 cancer cells (**a**). The downregulation of p-AKT and MMP2 and upregulation of P21 are shown in a statistic graph (**b**), which indicating the inhibition of apoptosis and cell migrate. Data were collected from 3 independent experiments and only representative pictures are shown. (All data were analyzed by GraphPad Prism. Unpaired Student’s *t* test was used when comparing treatment group with control group within the same cell type. *p* < 0.05 was considered statistically significant (* *p* < 0.05, ** *p* < 0.01).

**Table 1 molecules-28-06165-t001:** The inhibition rate of isolated compounds against HCT 116 cells at 20 μg/mL.

Compound	Inhibition Rate	Compound	Inhibition Rate
**1**	96.1%	**9**	76.1%
**3**	11.7%	**10**	3.4%
**8**	100%	**11**	24.5%

**Table 2 molecules-28-06165-t002:** The IC_50_ values of compounds **1**, **8**, and **9** against HCT 116 cells (μM).

Compound	IC_50_
**1**	0.02 ± 0.003
**8**	20 ± 0.006
**9**	16.3 ± 0.003

## Data Availability

Data are contained within the article or Appendix A.

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
