# Peer review of "Isolation and Anticancer Progression Evaluation of the Chemical Constituents from Bridelia balansae Tutcher"

_molecules, 2023, doi:10.3390/molecules28166165_

Round 1
Reviewer 1 Report
Recommendation: Accept after minor revision.
Comments:
(1) Page 2. Line 77. All the chemical structures of compounds were elucidation by comparing the NMR and HRMS data with literatures. However the HRMS data were didn’t supply druing “4.3. Compound Isolation and Structure Identification”.
(2) Page 3. Figure 1. The stereochemical drawing of compounds 1-3 should follow the rule of IUPAC Recommendations 2006 (Pure Appl. Chem., 2006, 78(10), 1897–1970).
(3) Page 4. Table 1. The cytotoxicity of all isolated compounds were evaluated (line 87). Why only 6 of 14 compounds inhibition rate were exhibited in Table 1?
(4) Page 9. Line 292. The CCDC access number of crystal 1 should be supplied.
(5) Please check the structural elucidation and reference of 2 carefully. It should be deoxypodophyllotoxin by comparing the NMR data. (J. Nat. Prod. 1997, 60, 401-403; Chem. Pharm. Bull. 1985, 33, 5565-5567).
(6) Compound 3 should be polygamain by comparing the NMR data. (Phytochemistry 1995, 39, 417-422; Planta Medica 1985, 51, 271-272). And the description “yatein (3) has not been reported in nature previously” in the manuscript should be removed.
Author Response
Dear reviewer,
Please kindly refer to the attachment uploaded. The revisions are made in blued or purpled font. Thank you!

Reviewer 2 Report
Zhao et al. have isolated compounds from Bridelia balansae Tutcher and evaluated their bioactivity. The work is valuable in context of identification of the group of compounds which are first reported in this plant although no novel compound has been identified. By the way, the work seems needing the following issues to be addressed:
Authors have screened a very limite bioactivity e. g. wound healing and anticancer effects; in that case; the title is suggested to be straight activity oriented
Authors have said they have made high throughput screening of 3000 plant extracts, does this work belong to all those 3000 plants. If not, that part needs to be removed to make more pin pointed.
The abstract should be revised based on the identity of the compounds as focused on the results section.
Why have the authors only x-ray crystallographically characterized only compound 1? Does it mena all others are done on the same aspect?
Line 63-66 seem the results are inserted in the introduction part, better to move those lines to results section.
Line 80, Please illustrate the approach how the chirality diversity has been measured. Have you analyzed the compound with chiral column? If so, what were the enantiomeric/racemic mixture compositions?
Line 87, adequate amount is not scientifically sound, you must use the exact amount (numerical value)
Line 109-110, the text formatting is not correct.
Figure 3b seems to show that the authors have compared the sample’s effect with that of DMSO, but the statistical difference is not indicated correctly.
Line 122, different concentrations is not correct phrasing, you have to use the concentrations.
Figure 6b, must show the annotation for the statistical differences.
Podophyllotoxin is randomly considered as microtubule destabilizing agent and antitumor agent. I think it is better to make uniform although functions are related. Additionally, Podophyllotoxin is a backbone of etoposide, teniposide the first anticancer precursor. Therefore, it must be introduced carefully.
The authors have assayed the wound healing effect but no interpretation on this effect in the discussion part. Even the whole discussion suffers from building analogical and critical insights on the results.
In the materials and methods section, authors have used a paragraph as General, but their numbers of reagents and solvents are unspecified in context of their purity/percentage/grade/source etc.
The sample was collected 9 years back, and data are presented now. How was the data procured? How were the samples preserved? I see nothing is said about sample standardization.
Line 222-223, please see the comment in the PDF file.
Line 311, degree Celsius is not written properly.
Line 325, different concentrations should be replaced by exact concentrations.
What was the standard used to compare the wound healing effects?
I think the conclusion section lacks the setbacks of the study, however there are many setbacks; Additionally, the final conclusion should be result-oriented.
Finally what makes you to choose these biological activities? Why not others?

Please see the attachment
Author Response

(The authors gave the same response as above.)

Reviewer 3 Report
In this manuscript, the authors report the isolation of a number of known phenolic natural products from Bridelia balansae. While no new compounds are reported, one compound is reported here for the first time as a natural product. The strength of the paper rests in the characterization of the anticancer activities of 4'-demethyl-4-deoxypodophyllotoxin; however, there are serious issues with this data, and other shortcomings in methods, that must be addressed prior to publication of this work. Overall, once these deficiencies are addressed, should be on interest of the readers of this special issue of Molecules.
This manuscript does not meet the standards of Molecules in terms of experimental detail, and should not be published until such detail is provided. Specifically:
Line 226: There is no information provided on how compounds 1-14 were isolated, other than which of these came from which crude fraction of the ethyl acetate extract in a rough outline provided in the supporting information. The authors must provide the detailed methods used to isolate pure compounds from these complex mixtures in the experimental section.
Line 231: Compound 1 is reported a crystalline, but no mp is reported. Authors should measure and report the mp of the crystals of 1. Also, no specific rotation is reported for 1. This is critical, as without this information there is no way to relate the reported absolute stereochemistry to that of other isolations of this compound from other sources.
Line 296: The CIF file for the x-ray structure determination of 1 must be provided as supporting information and this structure deposited with CCDC and FIZ Karlsruhe. Although the Flack x parameter is small, the uncertainty is large, indicating that this assigned absolute stereochemistry is not well defined.
Figure 6: There is no statistical analysis done on this data. This must be done before the authors can make any assertion of the effect of compound 1 on the levels of MMP2, p-AKT, or P21. There is no indication in the experimental section on the number of replicates for these Western blots. This information must be reported both in the experimental section and in the Figure legend.
There are a number of cases where no citation is provided. These references must be given:
Line 57: Missing reference for the HTS and IC50 values reported
Line 60: Need reference for statement on folk medicine use. Also here, the meaning of cataclasis is not clear: Is this supposed to be catacleisis, as in a palpebral adhesion?
Line 95: no citation is provided for the statement that previous work has demonstrate G2/M arrest in HCT-116 cells (but reference is provided on line 179).
Author Response

(The authors gave the same response as above.)

Round 2
Reviewer 2 Report
Authors have addressed the suggestions and recommendations